# The Effect of Exposure to Alcohol Media Content on Young People’s Alcohol Use: A Qualitative Systematic Review and Meta-Synthesis

**DOI:** 10.3390/ijerph22071078

**Published:** 2025-07-06

**Authors:** Sophie Getliff, Alex B. Barker

**Affiliations:** College of Health, Psychology and Social Care, University of Derby, Kedleston Road, Derby DE22 1GB, UK; s.getliff@derby.ac.uk

**Keywords:** alcohol, marketing, media, social media, social identity

## Abstract

Alcohol harm continues to present a public health priority. Whilst we know that a relationship exists between exposure to content and alcohol initiation and use in young people, the mechanism behind this effect is not well understood. Using the social identity approach as a meaning-making lens, a systematic review of the qualitative literature and meta-synthesis was conducted using Medline (all years), Embase (all years), and PsycINFO (all years). The inclusion criteria included studies which qualitatively explored the effect of exposure to content or marketing in young people (aged < 26). Twenty-two articles were identified and included in the synthesis and assessed for bias using the Downe and Walsh checklist. Four themes were identified: normalisation of alcohol use, branding and identity, marketing strategies, and acting in identity congruence. A line of argument was constructed arguing that alcohol content and marketing are often targeted at and work through social identities and cultural norms to normalise alcohol use and lead to increased consumption through embedding content and marketing in culture. These findings have implications for stricter regulations around alcohol marketing and the protection of young people from alcohol content.

## 1. Introduction

Alcohol harm continues to present a public health priority due to continued morbidity and mortality [1], with alcohol-related death figures in the UK rising for a fourth year running in 2023, with 10,473 lives lost [2]. This morbidity and mortality is associated with an estimated cost of GBP 6 billion to the National Health Service and a substantially greater cost to wider society [3]. Since alcohol consumption in adolescence is associated with a higher risk of consumption in adulthood [4], and therefore increased risk to health due to higher alcohol consumption [5], it is important to prevent children and adolescents from experimenting with these behaviours.

There is now strong evidence that exposure to, physically viewing, hearing, or being aware of advertising or other alcohol audio-visual content (AVC) in the media leads to alcohol experimentation in young people who do not currently drink and continued consumption in those who do [6]. There is also evidence that alcohol marketing, in any form, acts a behavioural trigger to drink in those at risk of an alcohol use disorder or those in recovery [7]. Whilst we know that the link between exposure and use exists, the mechanism, namely the psychological processes, behind such an effect has received little attention [8], with current explanations involving social norms and social learning theory [9], with the belief that adolescents copy influential others and celebrities through media exposure [10]. However, this does not explain how alcohol branding has a similar effect [11,12,13], with a suggestion that this works through a connection to social events [13], suggesting that underlying social processes may act as a mechanism. In line with this, Haslam et al. [14] suggest that we move away from ‘exposure theories of addiction’ as they do not account for the social and psychological influences that contribute to onset and continued use. The social identity approach seeks to explain why, how, and when individuals feel, think, and act in group terms, exploring the influence of the social on the individual [15], and may present a more plausible explanation for the relationship between exposure and uptake. In line with social identity theory, a shared sense of social identity can lead individuals to behave in congruence with the group to maintain self-esteem gained from being a part of the group [16], findings which help to explain the link between online social identities and alcohol use in adolescents [15]. If alcohol is marketed to a particular group, group members may consume alcohol or certain brands of alcohol to behave in congruence with that social group. In line with this hypothesis, emerging evidence shows that this is an effective marketing tactic used by the alcohol industry to target particular groups and establish identity norms around alcohol use [17,18].

Qualitative studies can help to explore the mechanism behind the persuasive effects of alcohol marketing by exploring the subjective effect of exposure to alcohol marketing. However, whilst studies have explored exposure to alcohol content in various types of media, the role of social identity in relation to marketing has received little attention [8]. Interpretative review methods can be used to combine qualitative interpretations, allowing for a reconceptualization of the current literature and allowing us to explore the role of social identity in secondary data across the literature.

The current review aims to explore the effect of alcohol content in the media to explore potential mechanisms for the effect on alcohol use and consumption, using the social identity approach as a meaning-making lens. A qualitative interpretive stance will allow the current study to explore this from the lived experience of exposure to this content, drawing on the existing literature.

## 2. Materials and Methods

The meta-ethnographic approach was used [19], which involved employing three stages in data analysis. The first step involved a search for reciprocal themes, phrases, and metaphors occurring frequently across studies. The second stage involved consciously searching for refutational metaphors, themes, and patterns that did not correspond with emerging patterns. The final stage involved constructing a line of argument, a statement that can summarise and express the emerging patterns across the data from the studies included in the review. In this paper, we argue that an additional initial step is required which takes place throughout all the subsequent steps, namely reflection. According to the principles of reflexive thematic analysis [20], the process of coding is flexible and organic, encouraging the researcher to embrace reflexivity, subjectivity, and creativity as assets in knowledge production [21]. SG is a post-graduate researcher working as a research assistant on the current project. AB is a lecturer in Psychology, with a research career in exploring unhealthy commodity promotion in the media, including alcohol.

Based upon an initial literature search and the small number of studies exploring the topic, the inclusion criteria for the peer-reviewed studies included (i) the consideration of young (<26 years old) people’s’ experiences of exposure to alcohol audio-visual content (marketing or otherwise) in the media and (ii) the use of a qualitative methodology. All forms of marketing were explored to provide an overview of the effects of exposure to alcohol marketing across a range of media formats.

As with systematic reviews, the quality of included studies in a meta-synthesis will affect the findings [22]. A grading system frequently used in meta-syntheses was used to assess the quality of studies [23,24], which incorporated scoring eight quality domains for each paper, namely (1) scope and purpose, (2) design, (3) sampling strategy, (4) analysis (5) interpretation, (6) reflexivity, (7) ethical dimensions, and (8) relevance and transferability, with the 8 criteria broadly encapsulating the 4 domains of quality proposed by Lincoln & Guba [25] as alternatives for quantitatively orientated criteria. Details on how to score these domains are described in Walsh & Downe [23]; briefly, each paper is assumed at the outset to have a high rating, and this decreases with the reviewer’s interpretation of flaws found in each of the eight domains, providing a final letter-based rating. The rating criteria are as follows:A—No or few flaws: The study’s credibility, transferability, dependability, and confirmability are high.B—Some flaws, unlikely to affect the credibility, transferability, dependability, and/or confirmability of the study.C—Some flaws, which may affect the credibility, transferability, dependability, and/or confirmability of the study.D—Significant flaws, which are likely to affect the credibility, transferability, dependability, and/or confirmability of the study.

Only studies rated C or above were included in the meta-synthesis, in line with previous meta-syntheses [26].

To ensure a clear and consistent literature search, key concepts and themes were tabulated before searches were undertaken (see Table 1).

The literature search was conducted on the 5 March 2025. The following databases were searched: Medline (all years), Embase (all years), and PsycINFO (all years). The search returned 2981 records, among which 905 were duplicates. The remaining 2076 were screened by title and abstract for relevance. A total of 22 articles were deemed relevant and were included in this review (see Figure 1). Articles were chosen by the senior author (AB) and confirmed and checked by the second (SG) to ensure they are relevant to the meta-synthesis and the validity of the reflexive interpretation of themes and concepts. No further methods to identify studies were conducted. Rayyan [27] was used to manage the literature search and review.

The authors immersed themselves in each paper before themes, concepts, metaphors, and phrases (the data) were extracted using the method described by [19]. Data were reciprocally grouped based on the similarity of meaning for themes, concepts, metaphors, or phrases before overarching themes were constructed. Refutational data were then examined and explored further before a line of argument synthesis was developed.

## 3. Results

Information about included studies can be found in Table 2.

Four key themes were identified, illustrating how alcohol consumption is normalised through media exposure, social identities, and social norms, leading to increased alcohol consumption. The strategic use of marketing tactics resonates with young audiences, establishing cultural norms and identities around consuming alcohol.

Theme 1: Normalisation of Alcohol Consumption

Alcohol use becomes normalised in young people’s digital and social lives, largely due to its pervasive presence in media and social media. Alcohol use is portrayed as a normal everyday activity [42], shaping cultural norms around alcohol use in our society [28,32,35,40]. This is particularly true with social media, which creates “intoxigenic digital spaces” where young people are frequently exposed to alcohol-related content [35,38,49]. This constant exposure integrates drinking into youth culture, making it seem like a natural part of everyday social interactions. Alcohol-related content is often shared in the context of friendship and fun, subtly reinforcing that alcohol consumption is a normal aspect of socialising among young people and reinforcing norms around drinking with peer groups, leading to increased alcohol consumption [35,38,39,47,48,49].

Moreover, family dynamics and community attitudes also play a significant role in shaping adolescent drinking behaviours. Cultural norms within the family and community influence perceptions of drinking, depicting it as a socially acceptable behaviour or rite of passage [32]. Similarly, Torronen et al. [36] notes that social media amplifies the visibility of alcohol consumption, especially in celebratory contexts, creating a sense of social normalcy around drinking. Adolescents are increasingly exposed to alcohol use as part of the celebrations and social rituals that dominate their digital spaces, further reinforcing drinking behaviours and association with brands.

Marketing plays a role in normalising alcohol use through associating brands with social norms and identities and the prevalence of this content within young people’s environment and daily lives.

Theme 2: Branding and Identity Construction

Alcohol content and branding is shaped by, and shapes, young people’s social identities. Adolescents are at a period where they are constructing their identities as they approach young adulthood. During this period, alcohol is perceived as a tool for social bonding and exploring emerging identities [28,34] due to its culturally normalised use. Due to social norms and expectancies around alcohol content and specific brands, young people may engage in drinking behaviours and brands to express their emerging identities. As such, young people may engage with alcohol brands due to the brand’s connotations with the identity which they are trying to portray [38] and to gain social status [41]. Similarly, young people may interact with brands on social media to reflect, shape, and express their emerging identity to others [30,31,34,36,37,39,41]. In this way, young people can gain social approval from their perceived in-group [31]. Alcohol brands are often marketed as lifestyle choices [38] tied to gender, maturity, and specific cultural activities, such as sports events or music festivals, leading young consumers to align their choices with these associations. Alcohol branding is deeply intertwined with cultural and gendered narratives [39], and young people often use these brands to construct and project their identities online. Marketing may also be tied to aspirational identities, further allowing young people to shape their identities based on their affiliation with alcohol brands [41]. Additionally, young people use alcohol brands as tools for social differentiation, choosing brands that signal affiliation with specific social groups [30]. This process allows them to construct identities that align with cultural values or peer expectations. In line with the social identity approach, young people may be inclined to behave in accordance with these identities, leading to increased consumption of alcohol brands.

Peer influence

Peer influence is a critical factor in the shaping of adolescent drinking behaviours, as evidenced in several articles. Atkinson et al. [38] show that although young people may not directly engage with alcohol brands on social media, they are still exposed to brand-related content through peer interactions. These interactions create a social context for drinking, where certain brands signify maturity or inexperience with drinking, influencing the choices of young consumers to portray experience.

Peer pressure plays a significant role in adolescents’ drinking behaviours [32]. The desire to fit in and gain social acceptance often leads young people to align their drinking habits with perceived group norms. Social media serves as a platform where young people seek validation from their peers, with drinking-related content garnering likes, comments, and shares. This feedback loop reinforces the idea that alcohol consumption is a desirable and socially acceptable behaviour. The influence of peer interactions is further amplified by social media algorithms, which prioritise engaging content, thus increasing young people’s exposure to drinking-related posts. Young people feel a desire to present an idealised self-image online, often influencing behaviour to share carefully curated drinking content that aligns with socially accepted norms of fun and sociability, further normalising alcohol use amongst social groups online [31,36,47,49]. Young women, in particular, face pressure to present themselves as socially active while maintaining control over their drinking behaviour. This balance between being “fun” and “responsible” is a key element in the curated drinking identities they project on social media [31]. Young people may also be selective about who they share content with, choosing to portray moderate and social activities around drinking on open channels but displaying transgressive content with in-group peers to build social relationships or to appear cool or funny [34,35]. This choice around what to share and who to share it with portrays a carefully curated identity around alcohol use, further associating alcohol use with social identities.

Young people may learn about social norms regarding alcohol use from their peers through social networking posts, which in turn may influence behaviours due to positive expectancies around alcohol [47,48,49].

Theme 3: Marketing Strategies

Alcohol marketing often involves aspirational messaging, which is often linked to the social identities which people form around drinking or want to portray. Alcohol marketing may appeal to young audiences through the portrayal of adult status, social success, and fun [32,41,43,45,46], potentially linking brands to an aspirational image which young people may wish to portray. Similarly, alcohol marketing often portrays aspirational messages [30,41,43,44] related to gender or social belonging. This recognition leads young people to associate certain alcohol brands with positively perceived social identities and peer acceptance, demonstrating how branding influences their perceptions of alcohol and becomes a part of a young person’s identity, leading young people to drink to fulfil a social image and maintain self-esteem.

Gender and cultural norms significantly influence young people’s drinking behaviours and their perceptions of alcohol marketing. Alcohol adverts aired during sporting events, often linking alcohol with masculinity, success and social status, are more likely to influence men [37]. Similarly, alcohol advertising acts on and reinforces traditional gender roles, with men depicted as adventurous and dominant and women as attractive and nurturing [31,40,42,44].

Marketing strategies are central to the widespread promotion of alcohol among young people. Young people are consciously aware of marketing [37,41]; however, while young people may claim resistance to marketing, they still engage with alcohol brand content in subtle ways, such as through likes and shares amongst their peer group, becoming active participants in the curation and dissemination of content, highlighting the challenges in regulating social media marketing [33,34]. The participatory nature of social media marketing makes it harder to monitor and regulate, allowing alcohol brands to embed themselves into young people’s online environments in ways that traditional advertising cannot.

Theme 4: Behaviour and identity congruence

Young people engage in drinking behaviours according to cultural norms and identities. Exposure to marketing and norms around drinking in society affects expectancies around alcohol use and positive attitudes towards drinking and certain brands [42,45]. Media consumers internalise the message they receive through their environment and consume based upon these internalised expectations [44]. These social norms and marketing affect behaviour; individuals may mimic behaviours they see from their peers to gain social approval and reinforce group identity, expectations, and norms [28,29,31], and individuals may try brands which have been advertised to their group identity [37].

Line of Argument

In conclusion, alcohol content in the media exhibits a strong social pull to engage in behaviours. Young people often perceive alcohol use as a normalised behaviour based on social norms and their identities. Alcohol marketing often uses identity-based marketing, advertising products using aspirational, fun, or gendered messaging which links certain brands to social identities of consumers in society. Young people then behave in line with their social identities, consuming alcohol or certain brands as a perceived group norm, further perpetuating the link between alcohol and social identities. The normalisation of alcohol consumption through social media, the strategic use of branding to shape identities, peer influence, and the targeted marketing tactics of alcohol brands all contribute to the embedding of alcohol use in youth culture. These themes demonstrate how marketing practices are not only designed to promote alcohol but also target demographics and identities to shape social norms and behaviours, particularly among young people who are navigating their identities in both digital and physical spaces. The pervasive presence of alcohol in media, coupled with peer and social pressures, creates an environment where drinking is seen as an integral part of social life, with significant implications for young people’s health and well-being. Whilst a relationship exists between alcohol marketing and consumption, this review goes beyond exposure-based explanations by positing that alcohol content and marketing are often targeted at young people and work through social identities and cultural norms to normalise alcohol use and lead to increased consumption through embedding content and marketing in culture; this normalised use leads to more exposure to content and marketing in society (see Figure 2).

## 4. Discussion

The current review set out to explore potential mechanisms for the effect of alcohol marketing on alcohol use, using the social identity approach as a meaning-making lens. It demonstrates how the alcohol industry markets their product to certain demographics and by doing so reinforces alcohol use amongst different social identities.

Exposure to alcohol content and marketing increases experimentation and use, in line with the literature. Previous research has suggested an effect of social learning, in that young people may imitate the behaviour of influential others; however, the line of argument from the current review suggests exposure to alcohol content and alcohol marketing may normalise alcohol use by influencing social norms and social identities.

This review is in line with previous research, suggesting that a person’s social identity may be targeted to market alcohol, with previous research identifying gendered marketing from the alcohol industry [50]. The current review is also aligned with research suggesting that people may drink to maintain identification with their social groups and peers [28,32,33,48], presenting possible mechanisms of the effect of alcohol marketing. A greater understanding of these mechanisms could allow more effective prevention, identification, and treatment for those with or at risk of an alcohol use disorder by building resistance to alcohol marketing, potentially through the construction of non-drinking identities, or by reducing social norms around drinking. Stricter regulation of alcohol advertising could lead to this.

Regulations exist for alcohol marketing. In the UK, the Advertising Standards Authority regulates TV, radio, and non-broadcast advertising with a general rule that alcohol advertising should not be aimed at under 18s or be likely to appeal to them [51,52]. Ofcom regulates broadcast television and radio programmes and state that alcohol misuse must not be ‘condoned, encouraged or glamourised’ [53]; however, for video-on-demand programmes, there are only rules for alcohol product placement, where alcohol brands pay to appear in programmes, not alcohol content in itself—for example, a character holding a glass of wine [54]. Ofcom are also implementing the Online Harms Bill; however, this includes illegal content and does not include alcohol content [55]. While exposure to alcohol content is not consistently regulated in the UK, the World Health Organization’s SAFER initiative calls for comprehensive bans on alcohol marketing [56], and with the current regulatory landscape in the UK, with different levels of protection from alcohol content based on the media one is consuming, young people are likely to continue to be exposed to alcohol content through their media use. Exposure alone may lead to normalisation of alcohol content; however, the current review suggests that this may work through altering cultural norms and linking alcohol and brands with social identities. This could lead to increased efforts to regulate specific marketing tactics, such as gendered marketing tactics [50], and the association of alcohol with sports [11,57,58,59]. Stricter regulation of alcohol content could prevent the normalisation of alcohol use in our society. Exploring how alcohol use is normalised in our society and through specific social identities, and therefore exploring the mechanism through which exposure to alcohol content and marketing exerts its effect, could be an alternative route to reducing the efficacy of this content. Future studies should continue to explore this mechanism, potentially leading to public health campaigns and interventions to reduce the association between cultural norms, social identities, and alcohol use.

This review was limited by the amount of research on the effects of media exposure to alcohol content. Due to this, we grouped all media forms, marketing techniques, and age groups to provide an overview of the topic area. Further studies are needed to explore this in more detail, exploring different age ranges and potential differing effects from different media forms and marketing techniques. A limitation of the meta-synthesis approach is the use of secondary data. This meta-synthesis aimed to provide a reinterpretation of the subject under investigation; therefore, the search terms used were not systematic or exhaustive but developed to elicit the topic under investigation. The authors acknowledge that this could introduce bias in the number of articles considered for inclusion. Only peer-reviewed articles were included in this review; as such, grey literature was not included. Only qualitative studies were included in the current review; a more comprehensive review including quantitative studies could lead to more generalizability in the findings. The authors note that there is potential for publication bias in the included studies.

## 5. Conclusions

In conclusion, exposure to alcohol content in the media is a factor in experimenting with alcohol and continued use. This review extends this view to consider societal norms and social identities as a potential pathway through which alcohol marketing exerts its effect, allowing for future areas of research to explore ways to minimise the effect of alcohol marketing and reduce the morbidity and mortality associated with alcohol use.

## Figures and Tables

**Figure 1 ijerph-22-01078-f001:**
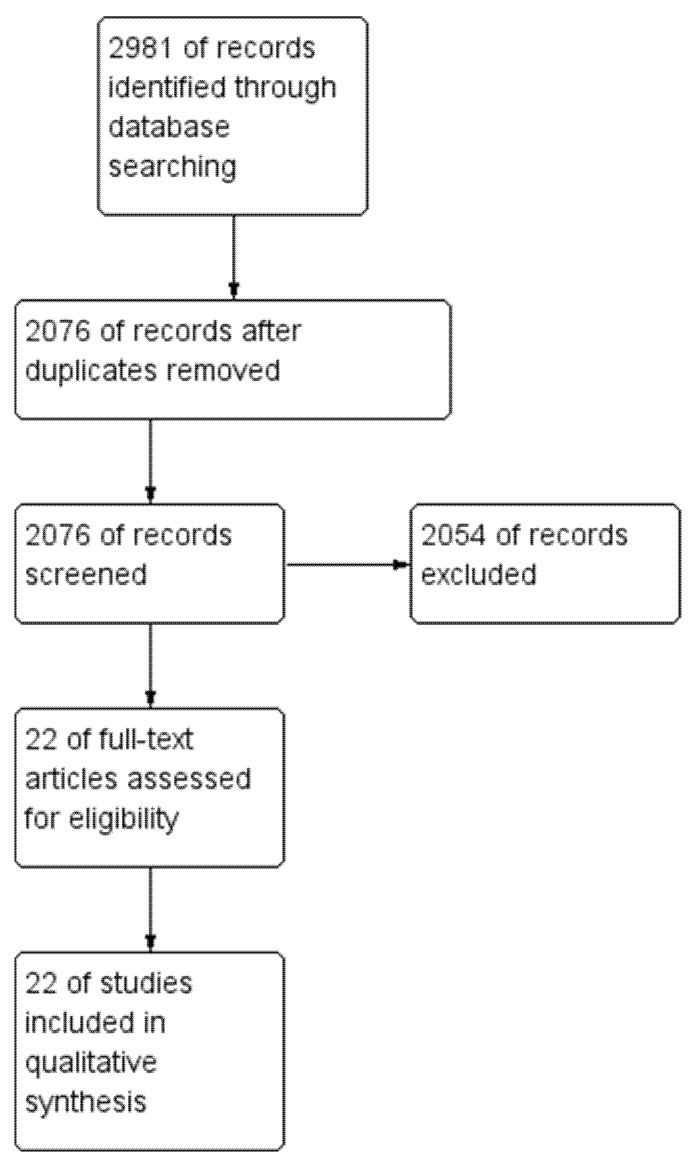
Flow diagram of included studies identified in the literature search.

**Figure 2 ijerph-22-01078-f002:**
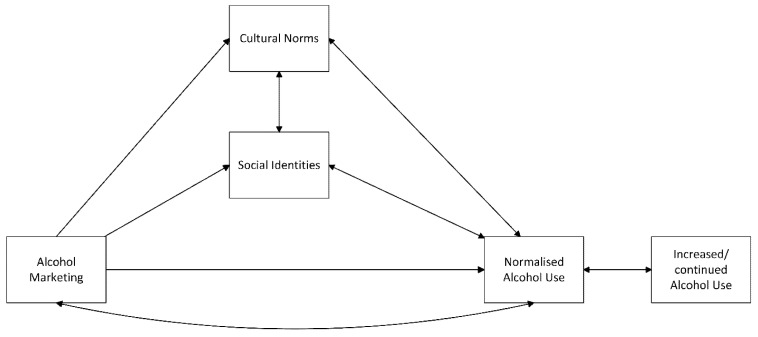
The social norms and identity model of exposure to alcohol content and marketing.

**Table 1 ijerph-22-01078-t001:** Search terms.

Concept	Adolescents	Alcohol	Media
Synonyms	Teenagers	Drinking	Mass Media
Broader	Young Adults Young People Youth	Ethanol	Entertainment
Narrower		Alcohol Use Alcohol Experimentation Alcohol Consumption Alcohol Misuse	Television TV Programmes Films Movies Cinema Digital Media Social Media
Related Terms		Intoxication Binge Drinking Alcoholics Alcoholism	Social Marketing Celebrity Endorsement Product Placement Advertising Marketing Influencer Sponsorship

**Table 2 ijerph-22-01078-t002:** Information and quality of included studies.

Study	Downe & Walsh Rating	Participants	Age Range	Exposure	Analysis	Country
Macarthur et al. 2020 [28]	A	42	14–18	Films, Social Media, Peer Influences	Thematic Analysis	England
Lyons et al. 2015 [29]	A	141 group discussions 23 individual interviews	18–25	Social Media	Thematic Analysis Foucauldian Discourse Analysis	New Zealand
Purves et al. 2018 [30]	B	48	14–17	Social Media	Thematic Analysis	Scotland
Atkinson & Sumnall. 2016 [31]	A	37	16–21	Social Media	Thematic Analysis	England
Scott et al. 2017 [32]	A	31	13–17	Any Marketing	Thematic Analysis	England
Niland et al. 2014 [33]	B	7	18–25	Social Media	Thematic Analysis	New Zealand
Steers et al. 2022 [34]	C	15	18–26	Social Media	Thematic Analysis	USA
Kaewpramkusol et al. 2019 [35]	C	72	20–24	Any Marketing, Social Media	Content analysis to identify themes	Thailand
Torronen et al. 2021 [36]	B	56	15–19	Social Media	Actor-Network Theory Analysis	Sweden
Dumbili & Williams 2017 [37]	B	31	19–23	TV Marketing, Films, Sports Sponsorship, Physical Marketing	Thematic Analysis	Nigeria
Atkinson et al. 2017 [38]	A	70	16–21	Social Media	Thematic Analysis	England
Niland et al. 2017 [39]	B	7	18–25	Social Media	Thematic Analysis	New Zealand
Dumbili & Williams 2016 [40]	B	31	19–23	All Marketing	Thematic Analysis	Nigeria
Gordon et al. 2010 [41]	C	64	13–15	All Marketing	Thematic Analysis	Unclear
Atkinson et al. 2013 [42]	B	114	11–18	TV Content and Marketing	Thematic Analysis	England
Waiters et al. 2001 [43]	C	97	9–15	TV Marketing	Categorical summaries	USA
Parker 1998 [44]	C	7	18–22	TV Marketing	Summary of patterns and themes	USA
Aitken et al. 1988 [45]	C	150	10–16	TV Marketing	Not mentioned	Scotland
Lieberman & Orlandi 1987 [46]	C	2766	11–12	TV Marketing	Content Analysis	USA
Vanherle 2025 [47]	A	32	15–18	Social Media	Thematic Analysis	Belgium
Merril et al. 2023 [48]	A	40	15–19	All Media	Thematic Analysis	USA
Corcorran et al. 2024 [49]	A	40	15–19	Social Media	Thematic Analysis	USA

## Data Availability

No new data were produced in the study.

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
