# Peer review of "The Effect of Exposure to Alcohol Media Content on Young People’s Alcohol Use: A Qualitative Systematic Review and Meta-Synthesis"

_ijerph, 2025, doi:10.3390/ijerph22071078_

Round 1
Reviewer 1 Report
Comments and Suggestions for Authors
This is an article about an extremely relevant subject – one that deserves this kind of insightful academic attention. I enjoyed reading it and I believe the suggestions I am putting forward can help improve its focus and widespread appeal.
General comments:
The subject of the analysis is clearly outlined. The methodology you used is adequate, although I have doubts about the some of the criteria adopted (cf detailed comments below). The bibliography is also adequate and updated. Generally speaking, the argumentation is clear, although some correction is required, at points (cf detailed comments below)
Detailed comments:
- 61-62: ‘explore … ‘explore’ – can this repetition be avoided?
- 77-79: is this information relevant here?
- 81-84: I have a few doubts about your inclusion criteria. I am not questioning the validity and the merit of your study, such as it is. However, to avoid being questioned on your motives for not establishing age cohorts or including every media + advertising, you should provide a thorough justification for deliberately doing so. Perhaps you could also tie it in with the ‘limitations’ (at the ‘Conclusions’) and the suggestions for further studies.
- 129: I didn’t understand this sentence (it might need expanding): if they are aware of the marketing (i.e., that this is marketing) they shouldn’t be influenced – at least, not so much, since they are aware they are being manipulated. This was how I understood it, but perhaps this is not what you meant. Please clarify. Also, I believe you retrieve this idea at a later stage. Perhaps you could focus totally on it here (or only later), in order to make it easier to follow your argument in toto.
(minor note: there are several occurrences: after ‘et al’, in references, there should be a plural, not a singular, since you are always referring to more than one author.)
- 161: …’maturity or inexperience’? I wasn’t very clear on what you mean here. Do you mean that certain brands can signify inexperience? Or is it the act of non-drinking that means you are inexperienced? Please clarify.
- 186: this idea of resistance to marketing is very relevant and should be further explored, if possible. Perhaps tie it with the observation on line 129, as per my previous comment?
In the ‘Results’ section, I found the argumentation slightly repetitive. I felt that you were returning to arguments which had already been put forward without much sense of progression. I believe this section requires a thorough re-reading and eventually some re-grouping of the arguments presented.
All in all, I do not know whether it was a good decision to group every age and every media/ad format. As it stands, I believe this has led to insufficient characterization of specific age groups (which could have been avoided had you focussed on one or two), and you were unable to adequately characterize the media/ad format at stake, which would have a bearing on the relationships audiences establish with them. However, as I said, this study still has merit, although this relative lack of focus, which is not explained, might account for this slight sense of circularity and repetition. As this point, as I pointed out before, the best way to overcome this, would be to provide a convincing, well-grounded justification for doing so.
- 253: Can you explain/expand? I could not follow what you mean here.
- 186: 18s (not 18’s)
- 262: just ‘misuse’, not use in general? Please clarify
- 264: product placement / alcohol content. I was not sure what you meant. What is the difference, specifically? Please clarify
- 280-281: I could understand this sentence. Please clarify
- 286: this information about peer-reviewing seems isolated and is not very relevant here. It should have been included in the inclusion criteria
- 280-291: why does this come before the conclusion?
l.293-297: The last paragraph of the previous section is repeated in the conclusion. It is too short for a concluding section. Limitations and suggestions for further research should come here, as well.
Observation: an important limitation is missing: you are not distinguishing between different age cohorts or diverse types of media/ads. Even if you can provide a convincing explanation for this option, it is still a limitation on the accuracy of the results you can get. You should acknowledge it and I suggest you can tie this in with (specific) suggestions for further studies, where this kind of differentiation would be factored in.
Comments on the Quality of English LanguageThere are some issues related to punctuation. These need to be revised.
Author Response
Dear Sir/Madam,
We thank the reviewers for their time in reviewing our manuscript and believe the suggested changes have greatly improved the manuscript. As suggested, we have been through and improved the accuracy of our writing throughout before addressing the reviewers' comments. Attached we have copied the reviewer comments (in red ink) and we have replied (in black ink).
We hope that you will agree that this manuscript has now been improved and is ready for publication.
Kind regards
Dr Alex Barker

Reviewer 2 Report
Comments and Suggestions for Authors
Thank you for the ability to review the manuscript, The effect of exposure to alcohol media content on young people’s alcohol use: a qualitative systematic review and meta-synthesis. While timely and interesting, I have made a few suggestions for improvement below:
Please use consistent language through the manuscript. For example, young adults/young people.
In the introduction, please define “exposure” and “mechanism”.
Can you also consider adding in prior research focusing on the marketing aspects and its effect on consumer behavior?
Lastly, please provide a better flow of discussion relating to the use of the theories in your study.
Please include the reliability measure of the screening and interpretation process (inter-rater reliability).
Furthermore, how was the coding process done? Again, please describe the validation of themes and inter-rater reliability.
Please clarify inclusion criterion: “<25 years” is mentioned, but the included studies include an age range up to 26.
In the results section, direct quotes and excerpts will help to add depth and validity to the themes extracted.
The theme of peer influence is under-developed. Consider integrating it more.
The model in Figure 2 should be better contextualized. What are the components? How do they interact? Consider explaining each component in-text.
The discussion section requires more connection to the theory. Consider explaining how social identity theory may guide future interventions or public health campaigns
The conclusion would benefit from clearer actionable recommendations—e.g., how should regulators or health educators use these insights?
Author Response

(The authors gave the same response as above.)

Round 2
Reviewer 2 Report
Comments and Suggestions for Authors
Thank you for updating the manuscript based on the feedback I provided. Here are a few minor concerns that need to be addressed:
You acknowledge the importance of this area and mention its inclusion in limitations for future research. This is acceptable, but a brief summary or citation of key prior studies would improve completeness even if it's outside the central scope.
While the manuscript now discusses implications for public health campaigns and policy, one or two hypothetical examples (e.g., “non-drinking identity” messaging via school-based or social media campaigns) would help translate the theory more concretely into application.
The limitations are mentioned but could be better organized into a distinct section (even a short paragraph) to improve visibility for readers. Consider adding the potential for publication bias in included studies and the limited generalizability due to the qualitative scope.
Author Response
Dear Sir/Madam,
We thank the reviewer for their latest round of feedback and have taken these considerations on board with the revised manuscript. Attached, we have copied the reviewer's comments (in red ink) and replied outlining our changes (in black ink).
We feel the manuscript is improved with these changes and hope the editor agrees that this is now ready for publication.
Kind regards
Alex Barker
